



# Significance of substrate soil moisture content for rockfall hazard assessment

Louise Mary Vick[1], Valerie Zimmer[2], Christopher White[3], Chris Massey[4], Tim Davies[5]

[1]Institute of Geosciences, UiT The Arctic University of Norway, Dramsveien 201, Tromsø 9009, Norway
5   [2]State Water Resources Control Board, 1001 I Street, Sacramento, California 95814, USA
[3]Resource Development Consultants Limited, 8/308 Queen Street East, Hastings, Hawkes Bay, New Zealand
[4]GNS Science, 1 Fairway Drive, Avalon 5010, New Zealand
[5]Department of Geological Sciences, University of Canterbury, Christchurch 8041, New Zealand

10   *Correspondence to*: Louise M. Vick (louise.m.vick@uit.no)

**ORCHID:** https://orcid.org/0000-0001-9159-071X

**Abstract.** Rockfall modelling is an essential tool for hazard analysis in steep terrain. Calibrating terrain parameters ensures that the model results accurately represent the site-specific hazard. Parameterizing rockfall models is challenging because rockfall runout is highly sensitive to initial conditions, rock shape, size and material properties, terrain morphology, and 15  terrain material properties. This contribution examines the mechanics of terrain scarring due to rockfall on the Port Hills of Christchurch, New Zealand. We use field-scale testing and laboratory direct-shear testing to quantify how the changing moisture content of the loessial soils can influence its strength from soft to hard, and vice versa.

We calibrate the three-dimensional rockfall model RAMMS by back analysing several well-documented rockfall events, adopting dry loessial soil conditions. We then test the calibrated "dry" model by adopting wet loessial soil conditions. The 20  calibrated dry model over-predicts the runout distance when wet loessial soil conditions are assumed. We hypothesis that this is because both the shear strength and stiffness of wet loess are reduced relative to the dry loess, resulting in a higher damping effect on boulder dynamics. For realistic and conservative rockfall modelling, the maximum credible hazard must be assumed; for rockfall on loess slopes, the maximum credible hazard occurs during dry soil conditions.

## 1 Introduction

25   The distribution of rockfall deposits is largely defined by topography, physical properties of the boulder (block shape, size, and geology), boulder dynamics (block velocity, rotations, bounce height, and impact and rebound angles), and substrate properties (Wyllie, 2014; Wyllie and Mah, 2004). Ground conditions will influence how much the kinetic energy of the block is reduced on impact with the substrate (Dorren, 2003; Evans and Hungr, 1993). A block impacting colluvial material or outcropping rock will retain much of its energy due to the stiffness of the surface. If the block impacts softer ground, some 30   of the block's kinetic energy will be dissipated as the soil deforms (Bozzolo and Pamini, 1986). Terrain parameters in soil





slopes will change seasonally, having a variable effect on rockfall runout behaviour; This is especially important for cohesive soils, where the changes in soil deformation behaviour in plastic and liquid states is significant.

In-situ rockfall experiments and other field data show that ground conditions have an influence on rockfall dynamics (Peng, 2000; Azzoni and de Freitas, 1995; Chau et al., 1998; Giani et al., 2004; Dorren et al., 2005; Ferrari et al., 2013; Volkwein et al., 2018). The analysis of block impact characteristics (Parronuzzi, 2009; Leine et al., 2014) allows for development of more realistic numerical simulation models. Within these models, terrain types must be accurately delineated and characterised for results to be meaningful (Dorren, 2003).

Terrain types need to be delineated according to the behaviour that most affects rockfall dynamics, by dividing substrate material into soft and hard portions. Hardness, the amount of plastic resistance to localised impact, will control how much energy is dissipated on boulder impact with the ground. We theorise that the hardness of soil is controlled by the shear strength and stiffness of the soil. These properties will have an effect on the dynamics of rockfall propagation. Where material shear strength and stiffness vary with soil moisture content, it is necessary to determine whether soils are dry or wet, and to assign specific "terrain" parameters to model the frictional forces that will be applied to a boulder during impact as it travels across them.

Discrete rockfall boulder runout events on the loessial soil slopes of the Port Hills, Christchurch, are affected by variations in soil moisture content (e.g., (Carey et al., 2017)), which can cause soil hardness to dramatically change their effect on rockfall runout. Constraining rockfall modelling parameters to reflect accurate rockfall behaviour requires characterising soil hardness changes due to moisture content.

In this paper, we analyse the results from two recorded rockfall events on loessial slopes in the Port Hills. Both sites have similar substrate material, slope gradient, roughness, aspect and density of vegetation. The three-dimensional rockfall model RAMMS was calibrated to a rockfall event (comprising the fall of multiple rocks) that occurred in dry conditions (Borella et al., 2016). The calibrated model was then tested by forecasting rockfall runout on the same slopes when the loessial soils were assumed to be wet. This was done to provide a data set and methodology for practitioners to apply when carrying out rockfall hazard and risk assessments under both wet and dry soil conditions.

## 2 Geological Setting

The Port Hills form part of Banks Peninsula, a volcanic edifice situated to the south east of Christchurch City (Figure 1). It was volcanically active in the mid-late Miocene, 11-5.8 Ma (Hampton and Cole, 2009). Hawaiian-style eruptions resulted in conical basaltic lava flow deposits radiating outwards from three principal eruptive centres and associated local vent structures (Brown and Weeber, 1992; Hampton and Cole, 2009; Hampton et al., 2012). An extended period of volcanic quiescence allowed widespread deposits of aeolian silt, the Banks Peninsula loess, to accumulate on the volcanically-formed slopes (Griffiths, 1973; Goldwater, 1990). These loessial soils are a product of pro-glacial fluvial action and wind transport/deposition (Davies, 2013); the dominantly quartz (>50%) and feldspar (>20%) composition of the soil reflects the



schist-greywacke mineralogy of the Southern Alps (Griffiths, 1973; Claridge and Campbell, 1987; Bell and Trangmar, 1987).

Post-depositional slope processes have resulted in reworking of the loess and loose volcanic material to form colluvium on the lower slopes, reaching 40 m thick in some foot-slope locations (Mcdowell, 1989; Jowett, 1995; Claridge and Campbell, 1987). Close to the underlying basaltic bedrock, mixed loess-volcanic colluvium is often recognised in the regolith profile (Bell and Crampton, 1986; Bell and Trangmar, 1987).

## 2.1 Port Hills Rockfall

The Canterbury Earthquake Sequence (CES) of 2010-2011 on the previously unmapped Greendale and Christchurch fault traces to the west and east of Christchurch produced seismic moments up to Mw 7.1 and high peak ground accelerations (≥1 g,) (Holden, 2011; Cousins and McVerry, 2010; Bannister and Gledhill, 2012; Wood et al., 2010; Beavan et al., 2011; Fry et al., 2011a, 2011b; Kaiser et al., 2012). These large, shallow (<15 km) ruptures triggered large slope failures on the Port Hills, of which rockfalls were the most abundant type and posed the most risk (Massey et al., 2014b). More than 6,000 individual boulders were mobilised, many of which impacted houses and affected the livelihoods of people within the impacted area. Rockfall is most likely to occur in closely-jointed or weakly-cemented material on slopes of ≥40° (Keefer, 1984). The columnar jointed lava flows of the Port Hills are generally dominated by three to four joint sets (Brideau et al., 2012) which vary somewhat between sites, attributed to variations in the paleotopography (Massey et al., 2014b). Scoria layers are interbedded with lava in some sites, and these have more widely-spaced discontinuities than the lava (Massey et al., 2014b). Rockfall data were collected by a rapid-response group immediately following events of the CES and resulted in a repository of data including 5,719 boulder locations (Massey et al., 2014), with their associated earthquake event and volumes (Figure 1).

## 2.2 Geotechnical Properties of Loess

Loess is defined as a loosely-deposited aeolian soil of predominantly silt-sized particles. Loess often displays high enough strength and cohesion to allow deposits to be meta-stable in a near-vertical exposure in dry conditions. When dry, the high cohesion of loess has been attributed to several possible mechanisms, including clay cohesion, calcite bonding, and soil suction (e.g. (Goldwater, 1990)). Post-depositional flocculation of cohesive clay grains to the larger silt- and sand-sized grains cause the irregular formation of clay 'bridges' between larger grains. As the larger grains within the soil do not touch, the mechanical behaviour of the material is dominated by the bonds between the larger grains (Gao, 1988; Lutenegger and Hallberg, 1988; Derbyshire and Mellors, 1988). Due to the cohesion between clay particles and negative pore pressure above any water table, loess generally displays a high dry shear strength; up to 180 kPa has been reported in Christchurch in loess of <10% moisture content (Mcdowell, 1989). However, loess has been observed to lose significant strength and cohesion





upon wetting, with cohesion and friction angle generally decreasing with increasing moisture content (Kie, 1988; Mcdowell, 1989; Della Pasqua et al., 2014; Carey et al., 2014). Wetting of the clay bridges and an increase in pore pressure reduces the shear strength of the material (Gao, 1988; Lutenegger and Hallberg, 1988; Derbyshire and Mellors, 1988; Della Pasqua et al., 2014; Carey et al., 2014).

The Port Hills loess is a cohesive predominantly silty soil with minor clay content. Strength parameters of the soil are largely controlled by the moisture content as repeatedly shown in testing (e.g. Tehrani, 1988; Mcdowell, 1989; Goldwater, 1990; White, 2016; Della Pasqua et al., 2014; Carey et al., 2014). A review of these studies (Massey et al., 2014a) shows that it displays high cohesion at moisture contents of <10%, while cohesion values are very sensitive to changes in the moisture content between 10 and 20% tests. Carey et al. (2014) found that at 3% moisture content the loess has cohesion of 45 kPa

and a friction angle of 48°. Comparatively at 16% moisture content displayed cohesion of 25 kPa and a friction angle of 28°. At moisture contents less than 15% the soil can display a brittle deformation style, the measured liquid limit for the Port Hills loess is a moisture content ranging from 22 to 28%, above which the material deforms as a fluid (Hughes, 2002).

## 3 Study Sites

Two Port Hills rockfall events are compared. The initial RAMMS model calibration at Rapaki Bay (Borella et al., 2016)

back-analyses mapped rockfall deposits from the 22[nd] February 2011 (NZST) earthquake. The calibrated model is then tested against data from a field experiment at Mt. Vernon conducted on 12[th] May 2014. Both slopes (which are within 0.6 km of each other, Figure 1) have similar gradient, aspect, and density of vegetation.

Rapaki Bay is a south-east-facing, moderately inclined (average 25°) slope with grass and tussock vegetation. The source area bedrock ranges from moderately to completely weathered basaltic lava and basaltic lava breccia, and the slope is

mantled by loess and loess-colluvium. The slope is situated above a small community; more than 200 boulders were released here during the 22[nd] February 2011 earthquake, impacting several houses. The slope falls from the peak (390 m asl) to sea level, with a c. 900 m-long runout zone, however all boulders stopped short of entering the sea.

Mt Vernon is a south-facing, moderately to steeply inclined (25-35°) slope in the Port Hills. Geology at Mt Vernon is similar to Rapaki Bay, outcropping bedrock also ranges from moderately to completely weathered basaltic lava and basaltic lava

breccia (again forming the rockfall sources). The slope is mantled by loess and loess-colluvium. The site was chosen due to its proximity to Rapaki Bay, its similarity in terms of materials, slope gradient, roughness and aspect, and low vegetation density, and because it has a safe zone for physical runout experiments. There is an obvious discontinuous rockfall source area above a well-constrained long (~700 m) runout zone and the uninhabited valley extends over 1.5 km from the boulder source areas to the nearest road, down slope.



## 4 Methods

### 4.1 Mapping at Rapaki Bay

Boulder deposit locations were measured in the field using a handheld GPS. Boulder size (lengths along three axes) and shape was recorded for most mapped boulders. New rockfall deposits were easily distinguished from paleo boulders by fresh
rock surfaces and location on top of the substrate. Impact scars on the substrate were mapped at both sites with lengths (axis parallel to boulder travel direction) and depths of 140 scars recorded. Additional mapped earthquake boulder data were contributed by the Port Hills Geotechnical Group - only boulder deposit locations were used from this data set. In total 336 boulders were mapped. To assess soil moisture conditions at the time of the earthquake, weather data were accessed through The National Climate Database (CliFlo, www.cliflo.niwa.co.nz) from the Governors Bay station, 3.5 km south-west of the
site and also south-east-facing (Figure 1). Moisture content of the soil was not tested at the time and instead inferred from published testing of 14 Port Hills Loess samples in January and February 2013 and 2014 (Carey et al., 2014; Table 3 (rainfall data)).

### 4.3 Soil testing

Moisture content and direct shear tests were conducted on 36 disturbed hand auger and borehole samples of Port Hills
loess/loess colluvium from 17 Ramahana Road and Centaurus Park (figure 1). Samples were taken from a range of soil profile depths (Table 1), and as such reflect a range of clay and natural moisture content and therefore mechanical properties. Testing was in accordance with *ISO/TS 17892-10:2004 Direct shear tests* and *NZS 4402:1996 Test 2.1 Determination of the water content*. Samples selected displayed a spread of both clay contents (Table 1; 5-19%) and natural moisture contents below, near, and above their 16-19% plastic limit (Table 1; 8-22%). The samples were subjected to 20kg, 50kg, and 100kg
applied weight (corresponding to 26, 64 and 126 kPa normal stress and overburden depths of 1.45 m, 3.64 m, and 7.28 m respectively with consideration of the average sample density (1750 km/m$^3$) and sheared at a constant rate.

### 4.4 Rockfall experiment Mt Vernon

Anthropogenic rockfall experiments involved the triggering and recording of 20 boulders at Mt Vernon. The boulders were jacked from the bedrock along cooling joints by inflation of air compression bladders. Each boulder was measured for size
and shape, dislodged, captured by video during travel, and impact trail (lines of impact scars) and deposit location were mapped. Locations were recorded with a handheld GPS and dGNSS. Seventy deposited boulder locations were mapped, including pieces from rockfall fragmentation. Nineteen impact scars were mapped and measured.
Thirteen soil samples were taken at Mt Vernon at the time of testing and analysed according to *NZS 4402:1996 Test 2.1 Determination of the water content* to obtain the natural moisture content.



### 4.5 Rockfall Modelling Approach

RAMMS::Rockfall, is a rigid-body three-dimensional rockfall simulation programme (Leine et al., 2013). It was chosen as an appropriate tool because: 1) it allows the user to create a boulder population of varying sizes and shapes modelled on point clouds of real boulders, and 2) the parameters that control different aspects of the terrain-boulder interaction process can be sensitively adjusted by the user.

In conventional rockfall models, rock interaction with the substrate is represented by coefficients of restitution, a ratio that defines the change in velocity after impact in both normal and tangential directions (e.g. Volkwein et al., 2011). In RAMMS the process of boulder interaction with a substrate is represented as a function of 'slippage' through near-surface material, a complex interaction with the substrate that includes sliding of a block through material until maximum frictional resistance is reached, and angular momentum generated by contact forces cause the block to be launched from the ground (Glover, 2015; Leine et al., 2013). The slippage can be parameterised (Table 2) for hard surfaces (e.g. rock) by decreasing the distance over which impact occurs and its time duration, to better reflect the instantaneous rebound observed in rock-rock interactions.

A robust RAMMS calibration exercise was performed for the Rapaki Bay dataset (Borella et al., 2016; this paper), and checked against other dry conditions datasets generated from the same earthquake sequence in other locations on the Port Hills. The modelling inputs included a representative sample of 21 mapped boulders with real shapes and sizes, a 3 m DEM (2013 LiDAR) and a terrain map delineated by changes in ground cover (outcropping rock, loess-colluvium, and loess). Following a recent RAMMS update (Bartelt et al., 2016) this calibration exercise was repeated to confirm relevance of the results.

Modelling of Mt Vernon boulder runouts was performed using the dry calibrated parameters. A second set of parameters was created to reflect the wet soil conditions by modifying the original parameters to incorporate more soil damping as the boulder interacts with the soil (Table 2). Parameters were adjusted incrementally until satisfactory results were achieved. To represent wet conditions, the parameter $\kappa$ was decreased by 16% for loess colluvium and 54% for loess, to reflect the longer slip distance through the soil ($1/\kappa$ = impact length). The parameter $\beta$ was decreased by 16% for loess colluvium and 33% for loess, to reflect the longer impact time ($1/\beta$ = impact time). The $\mu$-values were lowered by 33% for both soil substrates to reflect the decreased friction applied to the boulder over the period of the impact. The drag coefficient was increased by 40% for both soils, to represent the general greater drag on the boulder due to decreased soil hardness,

Inputs to the model included a representative sample of 5 boulders, which were based on the measured size and shape of the boulders used in the field experiments. A 3 m DEM (derived from the 2013 LiDAR) was used as the basis for the simulations, and a terrain map delineating field mapped changes in ground cover (outcropping rock, loess-colluvium, and loess) was used to proportion the locations of the various terrain material types across the slope.





The boulder density for both modelling exercises was 2700 kg/m$^3$, based on laboratory density testing of similar rock (Mukhtar, 2014).

## 5 Results

### 5.1 Soil conditions

Soil moisture tests from the Mt Vernon site in May 2014 showed water contents of between 28-62%. A prolonged rainy period preceded the experiments, with rainfall totals of 267, 263 and 44 mm recorded in March, April and May, respectively (the average totally monthly rainfall recorded since 1989 at the same weather station is 125, 144 and 88 mm for March, April and May respectively, Table 3).

Testing conducted by (Carey et al., 2014) in January and February 2013 and 2014 (when recorded rainfall for December, January and February was 65, 46, 29 and 105, 33, 48 for each year respectively) showed moisture contents ranging from 3.5 to 11%. The Rapaki Bay rockfalls occurred during typical dry summer conditions, when rainfall totals of 58, 50 and 38 mm were recorded for December, January and February, respectively.

Low moisture content (<10%) of the loess resulted in high cohesion (>35 kPa) for all clay content variations (Figure 4). Increased moisture content correlated with decreased cohesion; samples with 16-17% moisture displayed cohesions of 6-16 kPa for all clay contents. Moisture contents of >19%, above the liquid limit of the soil, displayed <5 kPa for all % clay contents tested. The spread of the cohesion data is large (±14.5 kPa) for varying clay contents at lower moisture contents, noticeable (±5 kPa) for intermediate moisture content and low (±1.5 kPa) for high moisture content. High clay contents correspond to higher cohesion values at low and intermediate moisture content, but the effect of clay content is negligible at high moisture contents.

### 5.2 Impact scarring

Mapped impact scars in the soil display a wedge-like form, with a clear boulder penetration point at the upslope end and a widening outwards and downslope, and an area of compression (where soil has been compacted and pushed up) with some excavated and overturned soil on the downslope end (Figure 4). Impact scar dimensions at both sites when compared



(P=0.035) showed variation in minimum, average and maximum depth:length ratio; 0.125, 0.22, 0.43 at Rapaki Bay, and 0.05, 0.29 and 0.4 at Mt Vernon respectively (Figure 3). Scars that show a greater depth:length ratio achieve depth in a shorter space during slippage (Figure 4a & b).

## 5.3 Modelling

Modelling was performed at Rapaki Bay to ensure that results were the same/similar following RAMMS updates since the publication of the original calibration (Borella et al, 2015). The RAMMS simulation of boulders at Rapaki Bay still compares favourably with the runout envelope of mapped boulders (Figure 5). Mapped and simulated boulder distribution within the envelope was compared: the largest proportion of boulders from both data sets were deposited in the upper slope (33° shadow angle), and the middle-lower slope (26° shadow angle). Both data sets showed a maximum runout of to within
the 22° shadow zone. The distributions of the data sets were both constrained by lateral ridges and a creek at the toe. A large proportion of the boulders from both data sets were channelled into a gully running parallel with the slope direction.

A RAMMS simulation of Mt Vernon boulder motions was performed using the dry calibration parameters. The runout envelope of the simulated boulders compares unfavourably to the envelope from the experimental rockfall rolling (Figure 6).
Runout of the simulated boulders is 175 m further downslope. The topography is more constrained than Rapaki Bay, with a channelisation effect that means lateral dispersion wasn't large; however the simulated rockfall showed a divergence of boulder paths into a neighbouring gully, behaviour that was not observed during the field experiments.
An adjustment of parameters from the original values, to reflect wetter soil conditions (Table 4), resulted in a better match between the field-experiment and runout simulation envelopes (Figure 6).

**6 Discussion**

Typical natural moisture contents in the Port Hills range from 10 to 25% (Goldwater, 1990). The moisture content at the time of the 22nd February 2011 earthquake was likely between 3 and 11% (Carey et al., 2014), representative of dry soil conditions. Soil moisture contents at the time of the Mt Vernon experiments were between 28 and 62%, due to a prolonged period of heavy rainfall in the months preceding the experiments, weather typical of the autumn season, and thus are
representative of wet soil conditions. High moisture content of the Port Hills Loess correlates well with low cohesion/shear strength. By increasing moisture contents past the liquid limit of the soil, cohesion values decrease from as high as 65 kPa to 5 kPa or less for all samples tested, regardless of the recorded proportion of clay particles within the samples. The amount of clay has an influence over the strength (cohesion) of the soil when dry (8-11% moisture), but in wetter conditions (15-18% moisture) its influence is reduced. When wet (moisture contents of 19-22%, above the plastic limit) the influence of clay



content is indistinguishable, with cohesion values at or below 5 kPa. This is likely due to the increase in pore pressure reducing the strength of the particle bonds.

Impact scar morphology displays evidence of the impact process: the soil penetration point and ploughing movement of the boulder - pushing soil forward as it slides in a down-slope motion causing compression and shear - reaches a maximum friction and rotational momentum marking the downslope and widened end of the scar. Overturned soil at the downslope marks the exit point of the boulder from the soil profile. A comparison of depth versus length of impact scars for the two field sites (Figure 3) shows that there is a greater depth relative to length of scarring during the winter when soil is wet, compared to the summer when the soil is dry. As the measured soil moisture content at Mt Vernon was above its liquid limit

(measured minimum 28%), the lower shear strength results in earlier plastic deformation and higher strain on boulder impact. As a result, the boulder achieves higher penetration depth within the soil during the 'slippage' process.

RAMMS modelling at Mt Vernon, using parameters calibrated to the Rapaki Bay data set (dry conditions), show that runout distance is overestimated when compared with rockfall field runout experiments. Adjustments of some of the RAMMS

terrain parameters, to reflect the lowering of shear strength of the loess, results in a more favourable match between the actual and modelled runout. All impact scars recorded during mapping at Rapaki Bay and following rockfall experiments conducted at Mt Vernon show a morphology that confirms the efficacy of the 'slippage' model component in RAMMS (and parameterisation thereof), and adjustments to the parameters set to reflect changes in impact dynamics under different soil moisture contents (and therefore strength) is valid.

We propose that under rapid-loading stress conditions (boulder impact), the proportion of recoverable (elastic) deformation is lower and irrecoverable (inelastic) deformation is higher for wet soil than for dry soil. We also propose that in a soil impact scenario, the irrecoverable stress proportion of the soil deformation in wet conditions results in a greater impact depth in the soil by the boulder due to lower stiffness, as noted by the increase in impact scar depth in wet conditions. Furthermore,

the greater plastic or viscous soil deformation under boulder impact loading in wet conditions results in a greater proportion of energy lost to the soil. As boulder motion in rockfall events ends when the kinetic energy is completely dissipated, the runout distance of the boulder will be shorter under wet soil conditions compared to the same soil under dry conditions.

By increasing the duration of slip through soil on impact in RAMMS, the decreased shear strength of the soil under wet

conditions is represented. The runout of dry vs wet soil modelling shows that by adjusting the parameters to suit the ground conditions, the actual runout is better represented. Dry soil will produce greater boulder runout distances than the same soil when wet. For hazard analysis purposes, practitioners should consider their terrain representation under different moisture

conditions within rockfall models to ensure the maximum possible rockfall runout and hence damage potential has been accounted for.

Representing soil conditions as only either dry or wet is a crude representation of a actual conditions. Realistically the mechanics of soil behaviour will change contiually with incremental increases in moisture content, and we recommend this contribution is further developed to explore the effect a range of moisture conditions will have on rockfall runout. In the future rockfall model parameterisation should be fine-tuned to a range of soil properties.

## Conclusions

Rockfall modelling using terrain parameters calibrated to rockfall events during dry loess soil conditions over-simulate
runout distance for rockfall events in wet conditions. Under wet conditions loess soil has a lower shear strength and depth of boulder penetration at impact during a rockfall event will be greater, resulting in a higher damping effect to the boulder and therefore a shorter overall runout distance. Rockfall model users should take soil conditions into account to ensure they have allowed for the worst-case runout distance when simulating rockfall events for hazard prediction purposes.

## Author contribution

Louise Vick- Conceptualisation, investigation, data curation, formal analysis, visualisation, writing- original draft, writing- review and editing,

Valerie Zimmer- Data curation, formal analysis, writing- original draft

Chris White- Investigation, formal analysis, writing- review and editing

Chris Massey- Funding acquisition, supervision, writing-review and editing

Tim Davies- Funding acquisition, invesitgation, validation, supervision, writing- review and editing

## Competing Interests

The authors declare that they have no conflict of interest.

## Acknowledgements

This project was undertaken at the University of Canterbury, and funded by the New Zealand Natural Hazards Research
Platform and GNS Science. Mt Vernon experiment costs were covered by Bell Geoconsulting Ltd with a contribution from Solutions 2 Access. The publication charges for this article have been funded by a grant from the publication fund of UiT



The Arctic University of Norway. We would like to acknowledge Marlène Villeneuve for help with editing, James Glover for assistance with the experiments, Marc Christen and the SLF Davos for assistance with RAMMS and providing multiple free licences, and Ellery Daines for edits and suggestions.

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

**Tables**

| Sample location | Sample depth (m bgl) | Clay content (%) | Moisture content (%) | Dry unit weight (kg/m3) | | | $\rho$ (newtons) | | | $\tau$ (kPa) | | |
|---|---|---|---|---|---|---|---|---|---|---|---|---|
| | | | | Test 1 | Test 2 | Test 3 | 20 kg applied weight | 50 kg applied weight | 100 kg applied weight | 26 kPa applied $\sigma_n$ | 64 kPa applied $\sigma_n$ | 126 kPa applied $\sigma_n$ |
| Hand Auger 5 | 1.0 | 18.5 | 9 | 1758 | 1709 | 1767 | 662 | 938 | 1293 | 84 | 119 | 165 |
| Hand Auger 1 | 2.0 | 18.4 | 17 | 1664 | 1731 | 1781 | 266 | 463 | 801 | 34 | 59 | 102 |
| Hand Auger 5 | 4.0 | 18.9 | 19 | 1689 | 1796 | 1775 | 171 | 374 | 801 | 22 | 48 | 88 |



| Hand Auger 4 | 1.0 | 15.4 | 9 | 1759 | 1788 | 1783 | 691 | 932 | 1414 | 88 | 119 | 180 |
| Hand Auger 3 | 2.0 | 15.3 | 17 | 1658 | 1663 | 1710 | 241 | 476 | 796 | 31 | 61 | 101 |
| Hand Auger 2 | 4.0 | 15.5 | 22 | 1666 | 1665 | 1667 | 201 | 407 | 807 | 26 | 52 | 103 |
| Hand Auger 4 | 3.0 | 10.2 | 10 | 1772 | 1822 | 1860 | 456 | 752 | 1145 | 58 | 96 | 146 |
| Borehole 3 | 2.8 | 7.6 | 8 | 1909 | 1949 | 1954 | 467 | 772 | 1209 | 59 | 98 | 154 |
| Borehole 1 | 7.0 | 8 | 16 | 1684 | 1724 | 1735 | 199 | 405 | 763 | 25 | 52 | 97 |
| Borehole 2 | 5.0 | 5.6 | 21 | 1719 | 1779 | 1779 | 202 | 52 | 101 | 26 | 52 | 101 |

**Table 1. Direct shear test variables for hand auger and borehole samples at various depths and displaying various moisture contents.**

| Parameter | Function |
|---|---|
| $\mu_{min}$ | Minimum sliding friction |
| $\mu_{max}$ | Maximum sliding friction |
| $\kappa$ | Time between $\mu_{min}$ and $\mu_{max}$ on contact with the ground |
| $\beta$ | Time between $\mu_{max}$ and $\mu_{min}$ as rock leaves the ground |
| *Drag coefficient* | Drag force applied to rock during ground contact |

**Table 2. RAMMS parameters used to define the slippage model**

| Series | Jan | Feb | Mar | Apr | May | Jun | Jul | Aug | Sep | Oct | Nov | Dec |
|---|---|---|---|---|---|---|---|---|---|---|---|---|
| 2010 | 66 | 23 | 30 | 24 | 216 | 205 | 64 | 175 | 58 | 49 | 63 | 58[1] |
| 2011 | 50[1] | 38[1] | 78 | 99 | 45 | 68 | 75 | 104 | 40 | 138 | 62 | 62 |
| 2012 | 48 | 70 | 54 | 38 | 26 | 92 | 110 | 207 | 54 | 103 | 78 | 65[2] |
| 2013 | 46[2] | 29[2] | 30 | 69 | 175 | 270 | 71 | 61 | 50 | 77 | 44 | 105[3] |
| 2014 | 33[3] | 48[3] | 267[4] | 263[4] | 44[4] | 53 | 84 | 41 | 41 | 36 | 85 | 30 |
| Average (1989-2018) | 55 | 49 | 63 | 79 | 103 | 106 | 93 | 107 | 67 | 73 | 60 | 65 |

**Table 3. Rainfall data (mm) recorded at the Governors Bay weather station in 2011, 2013 and 2014 with the weather station**

5 **average over 20 years provided for comparison.**

[1]**Rainfall preceding earthquake rockfall event, Rapaki Bay**
[2]**Rainfall preceding Carey et al (2014) testing, summer 2013**
[3]**Rainfall preceding Carey et al (2014) testing, summer 2014**
[4]**Rainfall preceding field experiments, Mt Vernon**

| Terrain | Calibration | $\mu$-min | $\mu$-max | $\beta$ | $\kappa$ | Drag layer coefficient |
|---|---|---|---|---|---|---|
| Outcropping rock | Original | 0.7 | 2 | 50 | 0.5 | 0.3 |
| Talus/colluvium | Original | 0.45 | 2 | 30 | 0.6 | 0.5 |
| | Wet soil conditions | 0.3 | 2 | 25 | 0.5 | 0.7 |
| Loess | Original | 0.3 | 2 | 30 | 0.65 | 0.5 |
| | Wet soil conditions | 0.2 | 2 | 20 | 0.3 | 0.7 |
| Asphalt | Original | 0.8 | 2 | 200 | 4 | 0.3 |



**Table 4: RAMMS terrain parameters (as described in Table 2) for typical Port Hills terrain types, calibrated to the original data set, and adjusted to wet soil conditions.**





**Figures**

**Figure 1: Location map of Christchurch and the Port Hills showing sites examined in this study. Red dots show mapped rockfall deposit locations (n=5,719).**




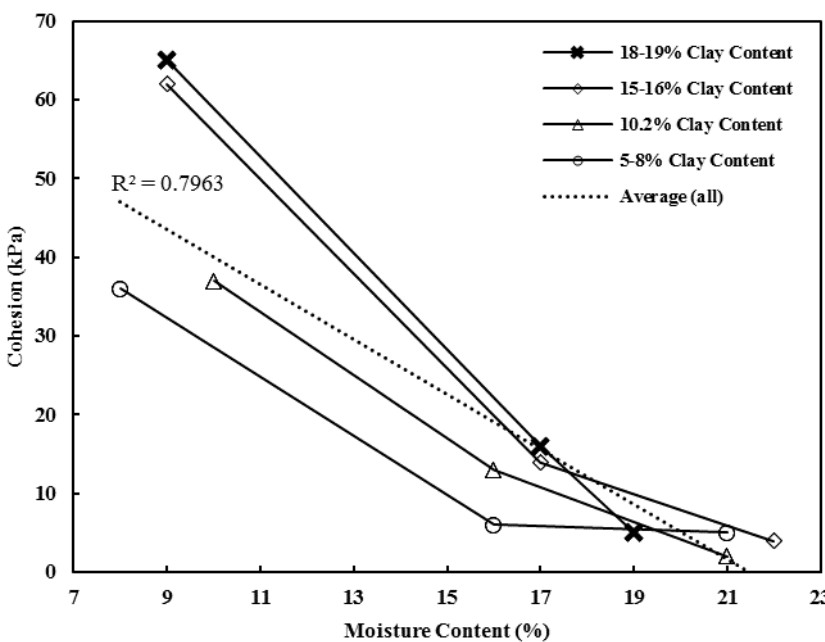

**Figure 2. Cohesion of loess at varying moisture contents, when loess clay content is varied.**

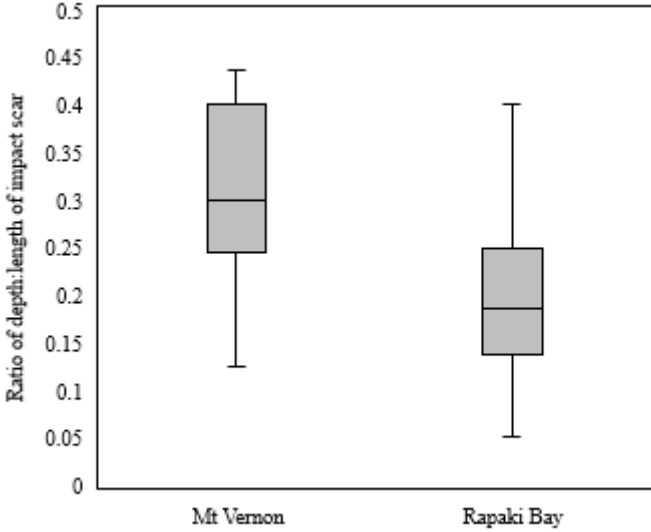

5     **Figure 3. Depth vs. length of impact scars measured at Rapaki Bay (n=140) and Mt Vernon (n=19). P=0.025.**



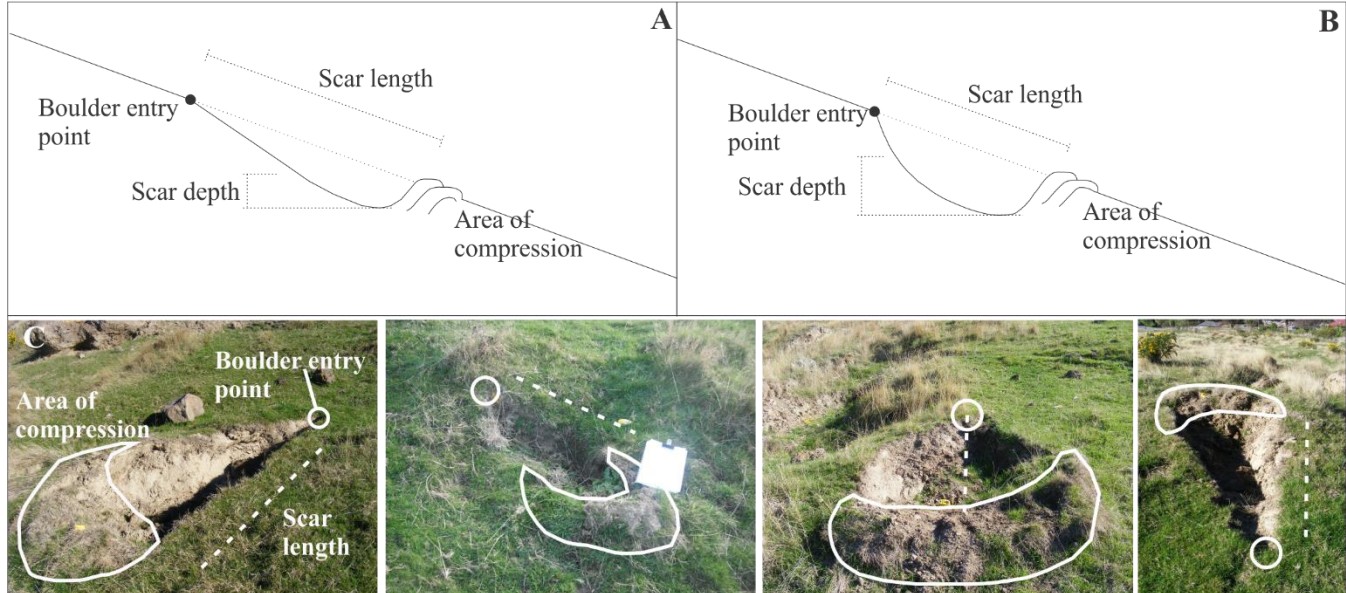

**Figure 4. Schematic representation of impact scar morphology, where depth:length of the scar ratio is low (a), representing dry conditions, and high, representing wet conditions (b). Images of impact scars from Rapaki Bay showing typical scar morphology from four different boulders (c).**





**Figure 5. Mapped (yellow squares, n=281) and simulated (purple circles, n=5292) rockfall boulder stopping locations within each shadow zone (the zone between projected shadow angles, after (Evans and Hungr, 1993)) at Rapaki Bay. Shadow zones are displayed from highest (darkest red=33°) to lowest (darkest green=22°). Runout extent of mapped (yellow line) and simulated (purple line) boulder populations are compared using envelopes. Inset: Proportion (%) of mapped and simulated boulders stopping within shadow zones.**

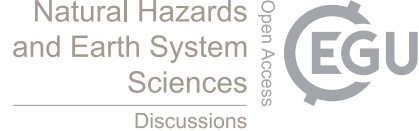

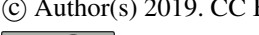

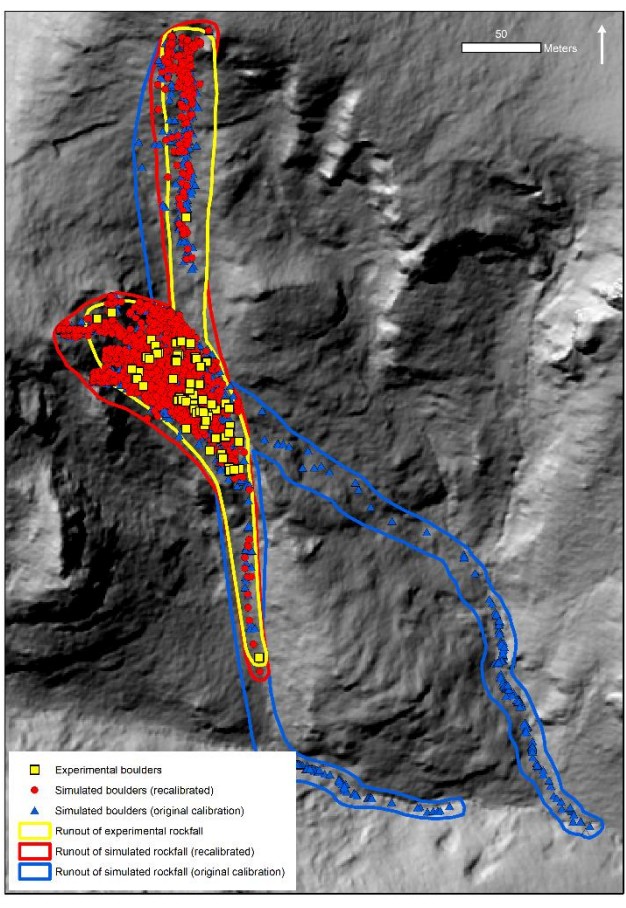

**Figure 6. Comparison of experimental rockfall (n=70) runout envelope (yellow line) with simulated rockfall using the initial calibration parameters (blue line = dry) and modified parameters (red line = wet) (boulder n=1800).**

