# Peer review of "Significance of substrate soil moisture content for rockfall hazard assessment"

_Natural Hazards and Earth System Sciences, 2019_

## Referee Comment (RC1) · Greg Stock (Referee) · 20 Mar 2019

This manuscript addresses how substrate soil moisture content affects the runout of rockfall boulders, with an emphasis on improving rockfall modeling and hazard assessment. This is the latest in a series of nice contributions utilizing data from rockfalls in the Port Hills, triggered by the 2010-2011 Canterbury Earthquake Sequence. Here the authors use field-scale testing and laboratory direct-shear testing to quantify how the strength of the local loessial soil changes with moisture content, and then evaluate those changes in cases of well-documented rockfalls. Their conclusion is that a certain soil will produce greater rockfall boulder runout distances when dry rather than when wet, because wet soil will dissipate more of a boulder's kinetic energy as the soil deforms. Although this finding not especially surprising, it is nice to see the re-
sults quantified in both field experiments and in model simulations. The results help to identify the maximum credible rockfall hazard, which occurs in dry soil conditions.

Overall the manuscript is well-written, balanced in its interpretations, and offers solid conclusions supported by data. I have a few suggestions for improvement below, but overall this is a valuable contribution to rockfall hazard assessment. My recommendation is to accept with minor revisions.

The amount of disparate data presented in the manuscript is impressive, and, at times, perhaps a tad overwhelming. These data include: (1) rockfall boulder runout from earthquake triggered rockfalls, in dry conditions (2) boulder runout data from experimental rockfalls, in wet conditions (3) measurements of boulder impact scars, (3) measurements of soil moisture content, (4) soil shear-test data, (5) models simulations of (1), and (6) model simulations of (2) in both wet and dry conditions. Because the manuscript addresses these data sources at two different study sites (Rapaki Bay and Mount Vernon), I had some trouble keeping track of the various conditions and results (e.g., remembering whether the rockfalls at Rapaki Bay happened under wet or dry soil conditions). I found it easiest to understand the results as a function of location, so I recommend that the authors make clear in each instance which site the results are from and what soil conditions that site represents. As an example, the caption for Figure 6 should state that these results are from the experimental rockfalls at Mount Vernon, which occurred under wet soil conditions. Ideally, it would have been nice to see additional experimental rockfall runouts from Mount Vernon in dry soil conditions to provide a direct comparison with the runouts in wet conditions, but I recognize that we cannot always perform science under ideal conditions!

The RAMMS rockfall model used in this study does not specifically address the conditions of the substrate that rockfall boulders impact, so the authors account for this by adjusting some of the RAMMS terrain parameters to reflect lowering of soil shear strength. The values that they adjust to account for wet conditions are reported in tables, but I found myself wanting more information on exactly how they derived the

adjusted values. Unless I missed it, the most information offered on this subject was the statement on Page 6, line 23: "Parameters were adjusted incrementally until satisfactory results were achieved." What does this mean, exactly? What defines "satisfactory"? The results may have been satisfactory, but the explanation of this methodology is not.

Along those lines, it might be useful if the authors discussed how other models using restitution coefficients to represent boulder impacts with the substrate could be modified to account for wet soil conditions; as is, the discussion is limited to the RAMMS model, which is only one of several rockfall runout models in use.

The sections on impact scarring somehow feel a little disconnected from the rest of the manuscript, even though they deal with a fundamental issue, namely how the effects of a wet or dry soil on rockfall impact are actually expressed in the field. Presumably the difference in the scar depth/length ratios at the two sites shown in Figure 3 is due to differences in soil conditions but this is not stated explicitly, either in the text or in the Figure 3 caption. This is relevant because Figure 3 provides evidence for the difference in the wet versus dry models results shown in Figure 6, where the dry conditions model significantly overestimated the actual rockfall runout. Perhaps the impact scarring data would feel more connected if the authors incorporated more discussion as to the usefulness of these measurements. As an aside: If soil conditions are not known at the time of a rockfall, could they be inferred from impact scar measurements, potentially offering a field-based method of soil characterization after the fact?

The authors tend to use passive voice (e.g., "samples were tested"), which leads to some ambiguity as to whether the authors performed certain analyses or whether they are referencing previous work. For example, on page 5, lines 10-13, it is unclear whether the authors inferred the moisture content of the soil at Rapaki Bay, or if this was done by Carey et al. (2014). Use of active voice (e.g., "we tested samples") can help to reduce ambiguity.

Regarding the rockfall experiment at Mount Vernon, the authors state that 20 boulders were triggered and mapped, yet figure the caption for Figure 6 indicates 70 experimental rockfall boulders. Why the discrepancy?

Figure 4 caption: The impact scars in "C" are representative of dry soil conditions (correct?), and thus only show examples of the schematic in panel "A". Are there similar photos of impact scars in the wet soil conditions at Mount Vernon? If so, it would be nice to show examples from both wet and dry conditions.

---

## Author Comment (AC1) · 28 Mar 2019

We thank Dr Stock for a well-balanced and considered manuscript review. His comments are clear and constructive and where possible all suggested changes to the text and figures have been made.

1. I found it easiest to understand the results as a function of location, so I recommend that the authors make clear in each instance which site the results are from and what soil conditions that site represents. As an example, the caption for Figure 6 should state that these results are from the experimental rockfalls at Mount Vernon, which occurred under wet soil conditions.

We have made these changes.

2. The values that they adjust to account for wet conditions are reported in tables, but I found myself wanting more information on exactly how they derived the adjusted values.

The explanation of the modelling method, when obtaining parameters to better reflect wet soil conditions, has been altered to the following (page 6, lower half): RAMMS parameters were adjusted incrementally until modelled runout paths showed a similar spatial distribution to that of the experimental boulder runouts. For each iteration of the model, parameters were adjusted to more closely represent wet conditions: the parameter $\kappa$ was decreased by 16% for loess colluvium and 54% for loess, to reflect the longer slip distance through the soil ($1/\kappa$ = impact length); the parameter $\beta$ was decreased by 16% for loess colluvium and 33% for loess, to reflect the longer impact time ($1/\beta$ = impact time); the $\mu$-values were lowered by 33% for both soil substrates to reflect the decreased friction applied to the boulder over the period of the impact; the drag coefficient was increased by 40% for both soils, to represent the general greater drag on the boulder due to decreased soil hardness. These adjustments to the parameters were considered suitable when the runout envelope of both the experimental rockfall and the modelled rockfall were closely aligned, rather than changing the parameters by a specific pre-determined value.

3. It might be useful if the authors discussed how other models using restitution coefficients to represent boulder impacts with the substrate could be modified to account for wet soil conditions; as is, the discussion is limited to the RAMMS model, which is only one of several rockfall runout models in use.

We have added the following lines to the discussion (page 9, lines 27-31): RAMMS is the only rockfall runout model currently available that represents boulder-substrate interaction as slippage, with parameterisation thereof. Other runout models may require a different approach to representing the change in soil conditions and its effect on the boulder runout, for example reduction of the tradition coefficient of restitution for wet soil conditions, to represent the increased damping effect the soil has on the boulder

during impact.

4. Perhaps the impact scarring data would feel more connected if the authors incorporated more discussion as to the usefulness of these measurements.

We have changed the caption and test of Figure 3 to clarify which impact scar depth:length ratios represent which moisture condition. The text has been changed in the results section (page 8, lines 1-5) to: Although both data sets display similar maximum depth:length ratios, the distribution of the values within the Mt Vernon data set (wet conditions) generally show a higher depth:length ratio. Scars that show a greater depth:length ratio are a result of impact of boulders which achieve depth in a shorter space during slippage/contact with the ground (Figure 4a & b). The Rapaki Bay impact scars show a generally lower depth:length ratio, indicative of shallower slippage through the soil during contact with the ground. The discussion section has also been altered to more clearly link the impact scar dimensions to soil moisture conditions and model parametisation (page 9, lines 5-17). Dr Stock raises an interesting question: If soil conditions are not known at the time of a rockfall, could they be inferred from impact scar measurements, potentially offering a field-based method of soil characterization after the fact? We interpret this question to be out of interest, rather than a suggested edit, and although we do not address this in text, agree that it is something that could be proposed as a future working direction. We would also suggest that future impact scar work link both boulder size and impact angle to scar morphology, as these likely have a marked effect on the resultant scar size and shape.

5. The authors tend to use passive voice (e.g., "samples were tested"), which leads to some ambiguity as to whether the authors performed certain analyses or whether they are referencing previous work. For example, on page 5, lines 10-13, it is unclear whether the authors inferred the moisture content of the soil at Rapaki Bay, or if this was done by Carey et al. (2014). Use of active voice (e.g., "we tested samples") can help to reduce ambiguity.

The use of passive voice has been altered where necessary to clarify which tests were conducted by the authors, and which test results are referenced from other published research.

6. Regarding the rockfall experiment at Mount Vernon, the authors state that 20 boulders were triggered and mapped, yet figure the caption for Figure 6 indicates 70 experimental rockfall boulders. Why the discrepancy?

We try to make it clear in the methods section- page 5 line 26- that the mapped boulder deposits at Mt Vernon include rockfall boulder fragments. The boulders generally fragmented on first impact, and tracking only one fragment would have been difficult. The test has been changed to: As most boulders fragmented on initial impact, all fragments were mapped as boulder deposits- therefore seventy deposited boulder locations were mapped, from the initial triggering of only 20 boulders.

7. Figure 4 caption: The impact scars in "C" are representative of dry soil conditions (correct?), and thus only show examples of the schematic in panel "A". Are there similar photos of impact scars in the wet soil conditions at Mount Vernon? If so, it would be nice to show examples from both wet and dry conditions.

Unfortunately, photos of scars from Mt Vernon in wet conditions are not good enough for level of quality required for published manuscript.

---

## Referee Comment (RC2) · Mark Eggers (Referee) · 22 Apr 2019

This is a great paper; a good story with some simple but excellent conclusions which are of significance to the assessment of rock fall. The manuscript itself could do with some revisions to add to the robustness of the paper. See the 'tracked changes' and comments in the attached supplement. In summary the main areas recommended for revision/extra discussion include:

1. Explanation of the term 'dry' in context of this study 2. Some figures showing the topography/slope morphology and mapped terrain types of the two sites would help characterise the physical setting of the study areas 3. Clarify where the Carey et al 2014 natural moisture content (NMC) data comes from relative to the Rapaki

[Figure]

Bay site especially with respect to physical setting so the reader can understand if these results are useful in making assumptions about natural moisture content at the time of the earthquake/rock fall event 4. Explain why sampling and testing was not carried out at Rapaki Bay and instead from another site on the other side of the hill 5. Explain why disturbed rather than undisturbed samples were used for the direct shear strength testing 6. Discuss the sample preparation (eg remoulding/recompaction etc) and testing (eg any pre-shearing etc) methods used for the direct shears 7. Discuss the limitations of the remoulded direct shears in assessing in-situ shear strength of loess 8. Given these limitations, while the change of shear strength with NMC trend appears to be a very reasonable finding, should some caution be noted in the paper about the correlation presented between the cohesion values obtained with NMC if there is some uncertainly about these results being representative of the in-situ shear strength? 9. Some simple graphs of the NMC test results and monthly rainfall data would help illustrate the differences and similarities across the sites

Please also note the supplement to this comment:
https://www.nat-hazards-earth-syst-sci-discuss.net/nhess-2019-11/nhess-2019-11-RC2-supplement.zip

---

## Author Comment (AC2) · 4 May 2019

We would like to thank Mr Mark Eggers for such a detailed and critical review of the manuscript. The suggested changes have increased the scientific standard of the manuscript greatly- it was especially helpful to follow the track changes. We have accepted most of the suggested changes to the text, and responded to the key points raised that require a more detailed response:

1.Why were the samples taken from this location on the other side of the hill ie the reader will want to know why the samples and testing wasn't done at Rapaki Bay. A sentence saying something like this would help: "Unfortunately no sub-surface investigations could be undertaken at Rapaki Bay. As such testing was carried out on

samples taken on similar soil types from a site investigation that was underway at the time of the study", or something like that. Were the samples/testing undertaken specifically for this study or were they part of a separate study (ie Chris White's thesis?). If part of another study and you are using the results you should state this and give a reference? By the way, if the Ramahana Rd and Centaurus Park sample sites are closer to the bottom of the hillslope where more colluvial soil content could be expected compared with Rapaki Bay/Mt Vernon (more upper to mid-slope??) could the grainsize distribution/clay content be different? Just trying to judge how relevant the Ramahana Rd and Centaurus Park test results are to the study sites on the other side of the hill.

Response: This amendment to the text has been added. Yes the soil sample locations were closer to the bottom of the hillslope, however we think that the range of clay contents within the samples shows the changes in mechanical behaviour depending on the clay content, and therefore reflects a range of different actual soil types. This is explain in the text (page 5, line 3-4): Samples were taken from a range of soil profile depths (Table 2), and as such reflect a range of clay and natural moisture content and therefore mechanical properties.

2. It would be useful to know where these 14 samples were taken relative to Rapaki Bay; so the reader can judge their relevance to helping make assumptions about NMC at Rapaki Bay at the time of the earthquake/rock fall event eg do the samples come from SE facing slopes as well? On a slope or on flat ground etc etc

Response: The samples were taken from the northern side of the hills (Lucas Lane, Maffeys Road, Redcliffs, Deans Head, Clifton Hill, Richmond Hill, Wakefield Avenue). This has been added to the text page 5 lines 13-14. Although it would be more ideal to have data from the southern aspect of the hills, as this is the only data that exists from the time period we have to make do.

3. I have a bit of an issue with use of the word 'dry' in this context. This applies throughout the paper. Dry to most people means no water or free from any moisture.

While no natural moisture contents (NMC) were tested at Rapaki Bay you have relied on the testing by Carey et al 2014 on samples taken during a similar time of year and similar monthly rainfalls. This testing shows NMC's were low but likely not totally without moisture (3-11% NMC from the Carey et al testing)? Perhaps when 'dry' is first used in the main text some context can be provided (see comment in last paragraph of intro below)?

Response: We agree with the comment, and in this context dry is used as an over-simplification of a sliding scale of behaviours. An amendment to the text has been added in the introduction, lines 11-12 page 2: In this paper the term 'dry' is used to indicate a soil with low natural moisture content, typically well below the plastic limit.

We have added typical atterberg limits of the soil at the end of page 3, start of page 4.

4. Think about two new figures showing the topo/slope morphology of each site eg hillshade maps of the lidar data as used as base maps in Figs 5 and 6 but without the other stuff over the top and with ground surface contours added.

Response: done, these are now figure 2.

5. You should probably elaborate on the test method used for the direct shear testing especially the procedures used for preparation and testing the samples. Given the samples were disturbed the testing must have been on remoulded material. So how was the material placed into the shear box/ring shear, in particular, how much compaction, any pre-shearing ie to simulate residual strengths given the sample is disturbed/remoulded etc etc. An issue with this testing is that the internal structure/fabric of the soil will be lost due to the disturbance. Given the importance of the internal structure of loess in-situ/undisturbed with regard to it's strength properties, does testing on disturbed samples give a realistic estimate of the shear strength changes with moisture content? I suggest you add a short discussion on the limitations of testing the shear strength of loess using disturbed, remoulded samples.

Response: The text has been added/rearranged to read as follows (page 5-6): Testing was in accordance with ISO/TS 17892-10:2004 Direct shear tests and NZS 4402:1996 Test 2.1 Determination of the water content. Samples selected displayed a spread of both clay contents (Table 1; 5-19%) and natural moisture contents below, near, and above their 16-19% plastic limit (Table 1; 8-22%). The samples were reconsolidated by means of tamping, using the Standard Procter test within the shear-box test sample rings. Twenty-five blows from the hammer were used to compact the soil directly into the shear-box test sample ring, and the method repeated with a fresh sample if the blows from the hammer caused the soil to be compacted to below or >5 mm above the height of the sample ring. The method was considered satisfactory, however there was an unavoidable amount of variation in the density of the samples: the dry density varied between 1658-1954 kg/m3, with an average of 1750 kg/m3. This variation can be attributed to the variable moisture contents of the soils that were compacted, which would have allowed greater or lesser compaction depending on the optimum moisture content for compaction, and the soil's particle-size distribution. The samples were subjected to 20kg, 50kg, and 100kg applied weight (corresponding to 26, 64 and 126 kPa normal stress and overburden depths of 1.45 m, 3.64 m, and 7.28 m respectively with consideration of the average sample density (1750 kg/m3)), and sheared at a constant rate.

The discussion (page 10 lines 13-20) has also been edited to read the following: The method of linking direct shear test results with soil performance under boulder impact is limiting, as the method of compacting disturbed soil during shear testing means that the internal structure of the soil is lost due to the remoulding. The strength values are therefore not wholly representative of in-situ conditions and greater accuracy in the strength properties of the loess would be achieved by performing similar tests on undisturbed samples. Furthermore, representing soil conditions as only either dry or wet is a crude representation of actual conditions. Realistically the mechanics of soil behaviour will change continually with incremental increases in moisture content, and we recommend this contribution is further developed to explore the effect a range of moisture

conditions will have on rockfall runout. In the future rockfall model parameterisation should be fine-tuned to a range of soil properties.

6.Any further details about the sampling? Where on the site/slope were the samples taken eg next to impact scars? How were the samples collected (small hand dug pit or hand auger?), what depth (especially relative to the depth of the impact scars) etc etc

Response: The text has been edited (page 6, lines 10-12) to read: Thirteen soil samples were taken at the time of the experiments and analysed according to NZS 4402:1996 Test 2.1 Determination of the water content to obtain the natural moisture content. Samples were collected as 30 cm tube samples from the base of 13 impact scars equally distributed down the slope.

7.Some simple graphs would really help here with understanding the soil test results eg plot the Mt Vernon NMC results against the testing by Carey et al which will help illustrate the differences between the two datasets. Secondly could plot the monthly rainfall data comparing the Dec-Feb 2013, 2014 data when the Caery et al samples were taken against the Dec 2010-Feb 2011 data when the earthquake/rock fall event occurred. If you do the plots then you can change the text discussing the compare-and-contrast without having to quote strings of numbers.

Response: These have been added as Figure 3a and b.